# Second Victim Phenomenon in an Austrian Hospital before the Implementation of the Systematic Collegial Help Program KoHi: A Descriptive Study

**DOI:** 10.3390/ijerph20031913

**Published:** 2023-01-20

**Authors:** Elisabeth Krommer, Miriam Ablöscher, Victoria Klemm, Christian Gatterer, Hannah Rösner, Reinhard Strametz, Wolfgang Huf, Brigitte Ettl

**Affiliations:** 1Karl Landsteiner Institute for Clinical Risk Management, Wolkersbergenstraße 1, 1130 Vienna, Austria; 2Wiesbaden Institute for Healthcare Economics and Patient Safety (WiHelP), Wiesbaden Business School, RheinMain UAS, Bleichstr. 44, 65183 Wiesbaden, Germany

**Keywords:** Second Victim, medical error, traumatization

## Abstract

(1) Background: The Second Victim Phenomenon (SVP) is widespread throughout health care institutions worldwide. Second Victims not only suffer emotional stress themselves; the SVP can also have a great financial and reputational impact on health care institutions. Therefore, we conducted a study (Kollegiale Hilfe I/KoHi I) in the Hietzing Clinic (KHI), located in Vienna, Austria, to find out how widespread the SVP was there. (2) Methods: The SeViD (Second Victims in Deutschland) questionnaire was used and given to 2800 employees of KHI, of which 966 filled it in anonymously. (3) Results: The SVP is prevalent at KHI (43% of the participants stated they at least once suffered from SVP), although less prevalent and pronounced than expected when compared to other studies conducted in German-speaking countries. There is still a need for action, however, to ensure a psychologically safer workspace and to further prevent health care workers at KHI from becoming psychologically traumatized.

## 1. Introduction

This article reports on the prevalence and symptoms of Second Victims in Clinic Hietzing (KHI), located in Vienna, Austria. This study was conducted using the SeViD (Second Victim in Deutschland) questionnaire published in March of 2021 [1] and also used in a modified version published in September of 2021 [2].

The term “Second Victim” was first described in the year 2000 by Albert Wu [3]. It refers to health care professionals who committed an error and are traumatized by it. This traumatization can lead to the manifestation of psychological and cognitive as well as physical reactions impacting the Second Victim negatively [3,4,5]. In 2009, the SVP was theoretically conceptualized by Susan Scott and was expanded to also include health care professionals who experienced an “unanticipated adverse event” and are traumatized by it [6]. This theoretical concept was further expanded in 2022 when an evidence- and consensus-based definition of Second Victim was published [7]: we refer to Second Victim as “any health care worker, directly or indirectly involved in an unanticipated adverse patient event, unintentional healthcare error, or patient injury and who becomes victimized in the sense that they are also negatively impacted”. Second Victims can develop dysfunctional coping strategies [3], possibly resulting in reduced quality of care for their future patients (possibly due to anxiety and fear of future errors increasing the future error rate) [8], them leaving their professions [9], or even committing suicide [10]. 

There have been studies within Europe [1,2,11], including Western Austria [12], discussing the prevalence and symptoms of the Second Victim Phenomenon (SVP). However, these studies have not investigated the prevalence and symptoms of Second Victims in one specific health care facility. Therefore, this study aimed to (1) conduct a descriptive study on SVP in one location (KHI); (2) compare the results of the study to similar previous studies; and (3) help develop targeted organizational strategies on how to manage the SVP in the future. 

## 2. Materials and Methods

This study is a cross-sectional, monocentric, descriptive study conducted before the implementation of the KoHi program at KHI. This institution has about 1000 beds for inpatients and 3000 employees. Approximately 52,000 inpatients and 250,000 outpatients are treated annually [13]. The questionnaire consisted of 14 questions and covered 6 dimensions (demographics, knowledge and exposure to SVP, the adverse event that led to traumatization, the recovery process, reactions, and support measures). Most items had to be answered on a 4-point Likert scale. One question gave the participants the opportunity to name the adverse event using free text.

The detailed construction and validation of the questionnaire used are described elsewhere [1,2,14].

The survey for this study was conducted from April to May of 2020 by handing out the paper forms to about 2800 employees of the KHI. Before the start of the survey, approval was given by hospital management and staff representatives. All employees were first informed verbally about the study by upper and lower management, which was briefed previously. A few days before the start of the study, all employees were informed once more via email. The survey was conducted anonymously, standardized, voluntarily, and in written form. All employees were given the survey including a cover letter. The completed questionnaires were put into opaque boxes, which were then collected by the house supervisor. Reminders were sent after two and four weeks. To avoid bias, the questionnaire was analyzed externally by scientific staff at the RheinMain University of Applied Sciences in Wiesbaden, Germany. The questionnaire was analyzed using the program evasys (evasys GmbH, Lüneburg/Germany). Percentages were calculated for questions one through twelve. The mean, standard deviation, and percentages were calculated for questions 13 and 14. Additionally, subgroup analyses of male vs. female, as well as nurses vs. rest were carried out, and the mean, standard deviation and percentages were calculated using Excel. Before conducting the study, an ethics application was submitted and approved (EK-19-074). 

## 3. Results

A total of 966 people participated in the survey.

As shown in Table 1, the majority of participants in the KoHi I study were female (79.3%). Most participants (53.1%) were very experienced, having worked in their fields for over 15 years in total, whilst 44.2% had worked at KHI for over 15 years. A total of 56.3% of the participants were nurses, including nursing assistants or nurses in training, while 13.3% of the participants were not part of the medical staff (listed in Table 1 as “Other occupations”). The vast majority of the participants (82.8%) directly performed patient care tasks in the course of their duties at KHI, while 7% worked indirectly in patient care (e.g., laboratory). A total of 5.1% of the participants had contact with patients in their work field but did not participate in patient care (e.g., facility management, cleaning services). Another 5.1% performed neither direct or indirect patient care tasks nor had any patient contact in their work context. In the male/female subgroup analysis, no significant differences were found concerning the demographics of the sample. When comparing nurses to the other occupational groups, 98.49% worked directly in patient care, whereas the other occupational groups only performed direct patient care tasks in 62.62% of the cases.

When asked about their knowledge of the term “Second Victim”, 50.6% of the participants stated they were familiar with it. Overall, 43% of the participants stated that they themselves were traumatized after an adverse event at least once. Of this 43%, 33.5% were traumatized within the last twelve months during their employment at KHI, while for 66.5%, the adverse event causing traumatization did not occur within the last twelve months.

Table 2 demonstrates the answers given to the question of which adverse event was the key experience leading the participants to become traumatized regardless of whether they themselves caused the event to happen or not. Most key experiences involved aggressive behavior from patients or their relatives (35.2%). For 30.1% of the participants, the key experience involved a patient unexpectedly dying or committing suicide. The following quotes describe some of the key experiences in more detail:


*“Accident: A young heavily pregnant woman was admitted after a car accident. Neither mother nor child could be saved”*



*“Attempted suicide of a patient”*



*“Extreme emergency/stressful situation during emergency shift”*



*“Death of a relative”*


After these traumatizing adverse events, 62.4% of the participants received help. Although, 6.2% of the participants have not received help despite having asked for it. In contrast, 31.3% of the participants have neither received nor asked for help. Those who have received help mainly received it, as shown in Table 3 below, from colleagues at KHI. No relevant differences were found in the male/female subgroup analysis. The analysis of nurses and other occupational groups showed that in 43.31% of the cases, aggressive behavior of patients or relatives caused the traumatization, whereas for the other occupational groups, only 19.31% were caused by aggressive behavior.

Overall, 12.9% of the participants who were traumatized after an adverse event recovered from that traumatization within less than a day. However, 31.7% needed a week, 25.1% recovered within a month, 14.6% recovered within a year, 7.7% needed more than a year, and 8% have not fully recovered yet. In all subgroup analyses, no significant differences were discovered.

Table 4 shows the answers given when asked about the extent of the participants’ reactions to the traumatizing adverse event. Table 5 refers to rating of helpfulness of suggested support measures.

## 4. Discussion

Being the first study to discuss the prevalence and symptoms of Second Victims in a German-speaking country for an entire hospital with about 1000 beds, this study shines a light on the specific problems and needs of the employees there and possibly in similar hospitals. Regarding the demographics of the participants, there are differences to other studies: while the SeViD I study concentrated on young (<=35 years) German physicians [1], the majority of participants in this study have many years of experience. In addition, this study does not concentrate on a certain occupational group but includes all occupational groups within KHI, though most of the participants were part of the nursing staff (56.3%). 

The study is not without limitations: of the 2800 employees surveyed, only 966 (34.5%) participated. Although this is a reasonable response rate for this kind of study, and the demographics of participants do not show systematic deviations from the overall demographics of the staff, as with any kind of voluntary study, convenience sample bias can occur. However, the comparably low incidents of Second Victim traumatization compared to previous studies [1,2] do not indicate an overestimation of the Second Victim prevalence. Of course, this study’s findings are not necessarily applicable to other hospitals because structures may vary. However, the main findings of this study do not deviate from previously conducted studies [1,2]. The risk of social response bias must also be noted, which is why this study was analyzed externally at the RheinMain University of Applied Sciences. A cross-sectional study’s findings, such as this one, always face limitations as mentioned above. Therefore, a follow-up study will be conducted in two years.

The knowledge of the term “Second Victim” (50.6%) was very high when compared to other studies in German-speaking countries [1,2,12], and was also slightly higher when compared to a study conducted by Edrees et al. at John Hopkins University (U.S.), where 46% of the participants were familiar with the term [15]. This can be interpreted as a positive development regarding the knowledge and acceptance of emotional or psychological distress (especially when traumatized during an adverse event) among health care workers. It also can be a sign of growing interest among health care workers for employee safety resulting in increased patient safety. KHI trained about 90 employees as patient safety experts including four lessons on the SVP before this study was conducted, which explains that many participants knew of the term.

International studies estimate the prevalence of Second Victims in health care at different levels. Scott et al. estimated approximately 30% of health care professionals to become Second Victims within the span of their work life [16], whilst a study by Wolf et al. conducted in 2000 suggested the prevalence to be 43.3%, though not being defined as the SVP at the time [17]. The SeViD I study estimated the overall prevalence of the SVP at 59% [1], while SeViD II estimated it at 60% [2]. In this study, the overall prevalence is estimated at 43%, ranking on the higher end when compared to international studies and on the lower end compared to studies in German-speaking countries. A possible explanation, when compared to the studies in German-speaking countries [1,2,12], is the long work experience of the participants leading to them being more resilient towards emotional stress or, when reflecting on events in the past, giving them lesser importance than they would have with less experience/resilience. This thesis is supported by a study conducted in 2021 including nursing staff, where age, work experience, and level of education had a significant positive correlation with nurses’ resilience [18]. Another study conducted in 2016, including health care workers in hospital and pre-hospital emergency settings, showed that more professional work experience was related to higher resilience levels [19]. Another possible explanation for the lower prevalence of Second Victims at KHI, when compared to the before-mentioned studies in German-speaking countries, could be that the employees at KHI are in general more sensitized to the SVP. This higher sensitivity speaks for a better-working climate regarding psychological safety at KHI in comparison to the other institutions involved in SeViD I [1] and II [2] and the study in Western Austria [12]. KHI offers crisis supervision and psychological counseling outside and inside of the hospital as well as pastoral care if needed by the employees. This might explain the comparably lower prevalence. Most key experiences leading to traumatization involved aggressive behavior of patients or relatives (35.2%) and unexpected death/suicide of a patient (30.1%). This result is similar to the findings of SeViD I [1] and SeViD II [2] when looking at the key incident of unexpected death/suicide of a patient (SeViD I = 35%; SeViD II = 29%). However, differences can be found when looking at the key experience of aggressive behavior of patients or relatives: in SeViD I, only 15% of the key experiences involved such behavior, whilst in SeViD II, the findings are similar where 25% of the key experiences involving the aggressive behavior of patients or relatives. SeViD I was conducted before COVID-19, and SeViD II, as well as this study, were conducted during COVID-19. A possible explanation could be that lockdowns and social isolation as well as heightened stress levels due to the pandemic caused rising aggression, which is supported by a study comparing aggression levels between people in lockdown and people under no such restrictions [20]. In addition, there could be differences between the probability to display aggressive behavior towards nurses (as in SeViD II all participants, and in this study, 56.3% of the participants being nurses) than doctors (SeViD I). In our subgroup analysis, nurses stated twice as much that their key experience was the aggressive behavior of patients or relatives. Reasons for that can be a lack of respect or social vulnerability, as suggested by Spence Laschinger [21], which aligns with the findings of our study.

Overall, 74.9% of the participants of this study stated that they received support from their colleagues at KHI after having suffered traumatization from an adverse event. This percentage is similar to the results of SeViD I (82% received support from colleagues) [1] but differs from the findings of SeViD II (49% received support from colleagues) [2]. As suggested before, concerning the lower prevalence of Second Victims at KHI, this could be due to a higher sensitivity for psychological problems and the employees having a better understanding and being more willing and capable to offer effective support to their peers. The importance of low-threshold peer support from colleagues is also stated in a study by Scott et al., where the first step of support for Second Victims is described as collegial help within their organization [6]. This already existing—but not yet organized—support net at KHI will be utilized in the future as a continuation of the KoHi I study.

Most Second Victims recover within a year after the traumatizing adverse event, as supported by different literature [1,2,22] and found in this study. Those who indicated they have not fully recovered yet (8%) need help to overcome their traumatization before developing dysfunctional coping strategies [3,9,23,24].

All but one (Support/Mentoring when continuing to work with patients) support measures were rated rather or very helpful on average. The top three measures were:The possibility to access legal consultation after a severe event;Access to professional counseling or psychological/psychiatric consultations (crisis intervention);Quick processing of the situation/quick crisis intervention (in a team or individually).

This shows parallels to findings of previous studies, which also support the support measure of the quick processing of the situation to be of higher importance than the other measures listed [1,2]. The possibility to access legal consultation after a severe event being rated the most helpful could indicate a fear of legal consequences. This can be seen as problematic for various reasons. Fear of (legal) consequences can reduce the willingness to report critical events and can also lead to health care professionals becoming over-cautious and, therefore, being more prone to committing errors, possibly creating a cycle of further traumatization. Further investigations about the possible need for training in legal advice may be useful.

At the start of the KoHi project, around 90 employees were already trained as experts in patient safety. This number will be increased within the next two years to offer more collegial support.

The results of this study justify the implementation of the KoHi support program. This program is based on other established Second Victim peer support programs, such as RISE at Johns Hopkins Hospital [25] and forYOU at the University of Missouri Health Care [16].

## 5. Conclusions

In this first hospital-wide survey in German-speaking hospitals, many employees were affected by the SVP showing that the SVP is a widespread problem. The symptoms described were highly relevant for health care workers and patient safety. The main requests were the possibility to access legal consultation as well as access to professional counseling and quick crisis intervention. Collegial support already plays a major role in dealing with Second Victim traumatization when compared to psychologic consultation or support from supervisors. Therefore, the establishment of a peer support program, such as KoHi, is justified.

## Figures and Tables

**Table 1 ijerph-20-01913-t001:** Characteristics of the participants.

Characteristics		Percentage of Participants
Gender	Male	20.7
	Female	79.3
Work experience (in total)	<1 year	6.6
	1–5 years	17.1
	6–10 years	12.9
	11–15 years	10.3
	>15 years	53.1
Work experience (at KHI)	<1 year	12.1
	1–5 years	19.8
	6–10 years	13.4
	11–15 years	10.5
	>15 years	44.2
Occupational Groups	Doctors (including interns)	12.6
	Nurses (including assistants and trainees)	56.3
	Other medical technicians	17.7
	Other occupations	13.3

**Table 2 ijerph-20-01913-t002:** Types of the most formative adverse event (key experience).

Type of Event	Percentage of Participants
Incident with patient harm	13.3
Incident without patient harm (near miss)	12.4
Unexpected death/suicide of a patient	30.1
Unexpected death/suicide of a colleague	6.5
Aggressive behavior of patients or relatives	35.2
Other	2.6

**Table 3 ijerph-20-01913-t003:** (Occupational) Groups which have helped the traumatized employees after an adverse event (multiple choice).

(Occupational) Group	Percentage of Participants
Colleagues at KHI	74.9
Directors	2.4
Counseling/Psychological consultation/ Psychotherapy	8.6
Superiors	35.5
Family/Friends	45.3

**Table 4 ijerph-20-01913-t004:** Extent of the reactions to the traumatizing adverse event.

Reactions	n	m (Mean)	SD (Standard Deviation)	Abstentions Total (%)
Fear of exclusion by colleagues	420	2.8	0.5	13 (3.1)
Fear of losing their jobs	414	2.7	0.6	14 (3.4)
Listlessness	410	2.5	0.7	16 (3.9)
Depressive mood	416	2.3	0.7	17 (4.1)
Concentration difficulties	415	2.3	0.7	10 (2.4)
Reliving the situation outside of professional life	405	2.4	0.8	24 (5.9)
Reliving the situation in similar professional situations	412	2.2	0.7	17 (4.1)
Aggressive, deliberately risky behavior	402	2.9	0.4	24 (6.0)
Defensive, overly cautious behavior	418	2.3	0.7	14 (3.3)
Psychosomatic reactions (e.g., head- or backaches, …)	410	2.2	0.8	19 (4.6)
Insomnia or excessive need for sleep	422	2.2	0.8	14 (3.3)
Use of alcohol/drugs because of the event	415	2.9	0.4	15 (3.6)
Feelings of shame	412	2.7	0.6	17 (4.1)
Feelings of guilt	420	2.4	0.7	13 (3.1)
Self-doubt	425	2.3	0.8	10 (2.4)
Social isolation	418	2.8	0.5	14 (3.3)
Anger towards others	424	2.5	0.7	9 (2.1)
Anger towards myself	417	2.6	0.6	10 (2.4)
Desire to be supported by others	397	2.1	0.8	31 (7.8)
Desire to process the event for better understanding	413	2.0	0.8	16 (3.9)

1 being the lowest and 3 being the highest possible rating (1 = very pronounced; 3 = not pronounced at all), the reactions were therefore rather less pronounced, the overall average rating being 2.46. The most pronounced reactions were the desire to process the event for better understanding and to be supported by others. The least pronounced reactions were the use of alcohol/drugs and aggressive, deliberately risky behavior. All subgroup analyses show no significant differences in these findings.

**Table 5 ijerph-20-01913-t005:** Rating of the helpfulness of suggested support measures.

Support Measure	n	m (Mean)	SD (Standard Deviation)	Abstentions Total (%)
The possibility to take time off from work directly to process the event	841	1.6	0.8	97 (11.5)
Access to professional counseling or psychological/psychiatric consultations (crisis intervention)	857	1.5	0.7	72 (8.4)
The possibility to discuss my emotional/ethical thoughts	848	1.7	0.7	78 (9.2)
Clear and timely information regarding the course of action after a serious event (e.g., damage analysis, error report)	856	1.7	0.8	70 (8.2)
Formal emotional support in the sense of organized collegial help	829	1.8	0.8	95 (11.5)
Informal emotional support	826	1.8	0.8	97 (11.7)
Quick processing of the situation/quick crisis intervention (in a team or individually)	866	1.5	0.7	62 (7.2)
Support/Mentoring when continuing to work with patients	821	2	0.8	97 (11.8)
Guidelines regarding the role/activities expected of me during a serious event	829	1.9	0.9	87 (10.5)
Support to be able to take an active role in the processing of the event	815	1.8	0.8	96 (11.8)
A secure possibility to give information on how to prevent similar events in the future	839	1.7	0.7	83 (9.9)
The possibility to access legal consultation after a severe event	848	1.3	0.6	71 (8.4)

For the question which support measures the participants believed to be more or less helpful, they ranked all suggested measures positively, meaning they expected them to be helpful (1 = very helpful; 4 = not at all helpful). They only ranked the possibility of support/mentoring when continuing to work with patients neutrally (m = 2), which was also, in conclusion, rated the least helpful support measure. The most helpful support measure rated by the participants was the possibility to access legal consultation after a severe event (m = 1.3), followed by the access to professional counseling or psychological/psychiatric consultations and the quick processing of the situation/quick crisis intervention in a team or individually (m = 1.5). All subgroup analyses showed no significant differences to these findings.

## Data Availability

The data presented in this study are available on request from the corresponding author.

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
