# Peer review of "Second Victim Phenomenon in an Austrian Hospital before the Implementation of the Systematic Collegial Help Program KoHi: A Descriptive Study"

_ijerph, 2023, doi:10.3390/ijerph20031913_

Round 1

Reviewer 1 Report

The methods for data collection by the authors are adequately described, as are the survey instruments used. The authors also explain the response rate (966) for the convenience sample and how it compares to other studies.

It is not clear to me the purpose of the study.  The aims of the study in the Abstract and the Introduction (Lines 41-42) do not align.  I am not sure what is meant by “shine a light”.  My sense based on the reading the results is that there are three aims: (1) conduct a descriptive study on SVP in one location; (2) compare the results of the study to similar previous studies; (3) propose/develop strategies for management of SVP in the future.

I have other issues with the presentation of data. Table 1 adequately described characteristics of the participants.  Is there any evidence that Males and Females differ in ways that would show differences in “formative adverse events” or in “reaction to the traumatizing adverse event”.

Another issue concerns the study results and discussion of other studies. The authors should first present the Table data only and other cross-tabulations such as with Male/Female in the Results section.  The Discussion section is a more proper place as to how the present study is similar/dissimilar with other studies. The authors clearly cited a number of other studies that are useful for the Discussion.  The content of the study increases our knowledge about SVP and how it might be mitigated.

Author Response

Reviewer 1

The methods for data collection by the authors are adequately described, as are the survey instruments used. The authors also explain the response rate (966) for the convenience sample and how it compares to other studies.

Thank you for your review and your valuable comments.

It is not clear to me the purpose of the study.  The aims of the study in the Abstract and the Introduction (Lines 41-42) do not align.  I am not sure what is meant by “shine a light”.  My sense based on the reading the results is that there are three aims: (1) conduct a descriptive study on SVP in one location; (2) compare the results of the study to similar previous studies; (3) propose/develop strategies for management of SVP in the future.

You are right, we should have specified the aims of the study more clearly. Therefore, we added the following paragraph:

Therefore, this study aimed to (1) conduct a descriptive study on SVP in one location (KHI); (2) compare the results of the study to similar previous studies and (3) to help develop targeted organizational strategies on how to manage the SVP in the future.

I have other issues with the presentation of data. Table 1 adequately described characteristics of the participants.  Is there any evidence that Males and Females differ in ways that would show differences in “formative adverse events” or in “reaction to the traumatizing adverse event”.

Thank you very much for this important input. We carried out subgroup analyses for gender (male/female) as well as for differences between occupational groups (nurses/other). All subgroup analyses showed no significant differences between the groups. We added this information in multiple sections of the manuscript, to make it more clear to the readers.

Another issue concerns the study results and discussion of other studies. The authors should first present the Table data only and other cross-tabulations such as with Male/Female in the Results section.  The Discussion section is a more proper place as to how the present study is similar/dissimilar with other studies. The authors clearly cited a number of other studies that are useful for the Discussion.  The content of the study increases our knowledge about SVP and how it might be mitigated.

Thank you very much for this valid point. We haven’t added cross-tabulations for males and females in the results section because no significant differences were found throughout our subgroup analysis. In alignment with your valid critique, we carefully revised the section concerning comparison with other studies that are only presented in the discussion section.

Reviewer 2 Report

Dear authors,

Thank you for the opportunity to review the paper ‘Prevalence and Symptoms of Second Victims in an Austrian Hospital before the Implementation of a Systematic Collegial Help Program (KoHi I Study).’

In my opinion, the research topic aligns well with the objectives and scope of the journal, as it covers occupational and patient safety issues as well as health behaviors of health professionals. However, several weaknesses exist, which are listed per section below. In particular, the introduction and the method need to be revised in a significant way.

Titel/Abstract

L1. In the title, the abbreviation KoHi I study can be omitted and replaced by the study type, e.g., descriptive study or cross sectional study/ survey.

The authors write that SCP is 'prevalent at KHI'. Here it would be interesting for the reader to know how high this prevalence actually is. Can you give a concrete number of prevalence here?

Introduction

L 42-43. The aim of this study is to ‘shed light on SVP in KHI and similar hospitals’. This could be better specified by the first sentence in the introduction (L. 23-27) and placed at the end of the introduction.

L. 34-44. What strategies are we talking about here? Are these meant for the organizational level?

Are second victims a phenomenon without theory? The introduction is lacking a proper theoretical foundation. What theory could be the basis for this phenomenon? What is the theoretical construct of the questionnaire based on?

Material and Methods

The methods section could start with a description of the specific type of study, e.g. monocentric, descriptive study. Then the authors could add the second sentence from the introduction (L. 27-29), which does not fit well in the beginning of the introduction.

How were the participants informed about the project/survey and who handed out the questionnaires?

It would be helpful if the authors could provide some information about the questionnaire used.

Which dimensions are specifically covered (one ore more?) and what are the questionnaire's psychometric criteria (objectivity, validity, reliability)?

How many items are included? How are the answers to the items (Likert rating scale?) and how long do the participants need to complete the questionnaire?

Which calculations were subsequently carried out (e.g., only descriptive analysis, percentages, mean, standard deviations, ..) with which program? SPSS, Excel?

Results

The authors also include qualitative results in the study (L. 85-92). This should be clear described in the methods section. Was there a possibility for an open response?

 Tables 4 and 5 could include a description of the abbreviations used in the tables.

What is the impact of abstention on the interpretation of the results?

Why were no subgroup analyses carried out? The results of the nurses in particular would be interesting, as they have the strongest contact with the patients and are overrepresented in the survey with almost 60%.

Conclusion

What can the authors recommend after finalizing this study? Which targeted strategies, as stated in the introduction (L. 43-44) should be developed? What about the result from L. 210 ‘1. The possibility to access legal consultation after a severe event’?

L. 234-235 What does KoHi mean?

Thank you very much!

***end of review***

Author Response

Reviewer 2

Dear authors,

Thank you for the opportunity to review the paper ‘Prevalence and Symptoms of Second Victims in an Austrian Hospital before the Implementation of a Systematic Collegial Help Program (KoHi I Study).’

In my opinion, the research topic aligns well with the objectives and scope of the journal, as it covers occupational and patient safety issues as well as health behaviors of health professionals. However, several weaknesses exist, which are listed per section below. In particular, the introduction and the method need to be revised in a significant way.

Titel/Abstract

L1. In the title, the abbreviation KoHi I study can be omitted and replaced by the study type, e.g., descriptive study or cross-sectional study/survey.

Thank you very much for your review. We added the study type as you suggested and the title now reads:

“Prevalence and Symptoms of Second Victims in an Austrian Hospital before the Implementation of the Systematic Collegial Help Program KoHi: A Descriptive Study”

The authors write that SCP is 'prevalent at KHI'. Here it would be interesting for the reader to know how high this prevalence actually is. Can you give a concrete number of prevalence here?

You are right, we did give the exact number in the abstract. Therefore, aligning with your suggestion, we added it. Thank you for this important input. It now reads:

The SVP is prevalent at KHI (43 % of the participants stated they at least once suffered from SVP), although less prevalent and pronounced than expected when compared to other studies conducted in German-speaking countries.

Introduction

L 42-43. The aim of this study is to ‘shed light on SVP in KHI and similar hospitals. This could be better specified by the first sentence in the introduction (L. 23-27) and placed at the end of the introduction.

Thank you for this comment. Of course, we should specify the aims more, as you suggested. We changed that part of the article and specified the aims, so it now reads:

“Therefore, this study aimed to (1) conduct a descriptive study on SVP in one location (KHI); (2) compare the results of the study to similar previous studies and (3) to help develop targeted organizational strategies on how to manage the SVP in the future.”

  1. 34-44. What strategies are we talking about here? Are these meant for the organizational level?

Thank you for your question. The study we report on in this article is the first of five to be conducted at KHI. KHI plans on training colleagues to give adequate peer support to Second Victims within the hospital. We therefore have added the information, that the strategies will be based on organizational strategies (see above).

Are second victims a phenomenon without theory? The introduction is lacking a proper theoretical foundation. What theory could be the basis for this phenomenon? What is the theoretical construct of the questionnaire based on?

Thank you for this valid point. We focused on the theoretical conceptualization by Susan Scott, who gave the SVP a theoretical foundation. Also, we added the very recently updated definition of the SVP that was consensus- and evidence-based by the European Researchers’ Network Working on Second Victims (ERNST). It now reads:

“The term “Second Victim” was first described in the year 2000 by Albert Wu [4]. It refers to health care professionals which committed an error and are traumatized by it. This traumatization can lead to the manifestation of psychological, cognitive as well as physical reactions impacting the Second Victim negatively [3–5]. In 2009 the SVP was theoretically conceptualized by Susan Scott and was expanded, now also including health care professionals who experienced an “unanticipated adverse event” and are traumatized by it [6]. This theoretical concept was further expanded in 2022, where an evidence and consensus-based definition of Second Victim was published [7]: We refer to Second Victim as “any health care worker, directly or indirectly involved in an unanticipated adverse patient event, unintentional healthcare error, or patient injury and who becomes victimized in the sense that they are also negatively impacted”.”

We also have extended the description of the SeViD-questionnaire, as construction and validation are described elsewhere.

Material and Methods

The methods section could start with a description of the specific type of study, e.g. monocentric, descriptive study. Then the authors could add the second sentence from the introduction (L. 27-29), which does not fit well in the beginning of the introduction.

You are absolutely right. Therefore, we deleted the second sentence from the introduction and added it to the methods section, as you suggested. It now reads:

“This study is a cross-sectional, monocentric, descriptive study, conducted before implementation of the KoHi program at KHI. This institution has about 1.000 beds for inpatients and 3.000 employees. Approximately 52.000 inpatients and 250.000 outpatients are treated annually.”

How were the participants informed about the project/survey and who handed out the questionnaires?

Thank you for this question. The participants were first informed verbally by their supervisors. The supervisors were informed by management. Then, shortly before the questionnaires were handed out, all employees received an email informing them that they would be sent the questionnaires for the KoHi I study. To also clarify this to the readers, we added this information to the manuscript. It now reads:

“All employees were first informed verbally about the study by upper and lower man-agement about the study, which was briefed previously. A few days before the start of the study, all employees were informed once more via email.”

It would be helpful if the authors could provide some information about the questionnaire used.

Which dimensions are specifically covered (one ore more?) and what are the questionnaire's psychometric criteria (objectivity, validity, reliability)?

We added the specific dimensions of this content-validated questionnaire. It now reads:

“The questionnaire consisted of 14 questions and covered six dimensions (demographics, knowledge and exposure to SVP, the adverse event that lead to traumatization, the recovery process, reactions, and support measures. Most items had to be answered in a 4-point likert scale. One question gave the participants the opportunity to name the adverse event in free-text.”

How many items are included? How are the answers to the items (Likert rating scale?) and how long do the participants need to complete the questionnaire?

Thanks for this question. We added the following paragraph to describe the questionnaire further:

“The questionnaire consisted of 14 questions and covered six dimensions (demographics, knowledge and exposure to SVP, the adverse event that lead to traumatization, the recovery process, reactions, and support measures. Most items had to be answered in a 4-point likert scale. One question gave the participants the opportunity to name the adverse event in free-text.”

Since the questionnaire was handed to the participants in paper form to take with them, the time it took for each participant to complete it was not measured. We assume it took less than five minutes to fill in the questionnaire, since it was only one double-sided sheet of paper.

Which calculations were subsequently carried out (e.g., only descriptive analysis, percentages, mean, standard deviations, ..) with which program? SPSS, Excel?

We added the following explanation:

“The questionnaire was analyzed using the program evasys (evasys GmbH, Lüneburg/Germany). Percentages were calculated for questions one through twelve. Mean, standard deviation and percentages were calculated for questions 13 and 14. Additionally, subgroup analyses of male vs. female, as well as nurses vs. rest were carried out and mean, standard deviation and percentages were calculated using Excel.”

Results

The authors also include qualitative results in the study (L. 85-92). This should be clear described in the methods section. Was there a possibility for an open response?

The questionnaire was described in more detail, as stated above. Thank you for this important input.

 Tables 4 and 5 could include a description of the abbreviations used in the tables.

A description was added to both tables as you suggested.

What is the impact of abstention on the interpretation of the results?

Thank you very much for this question. To add more transparency, we’ve added the percentage of abstentions. Most abstentions, however, were below 5 % only few above 10%. During our analyses we did not discover systematic differences to other questions with lower abstention rate.

Why were no subgroup analyses carried out? The results of the nurses in particular would be interesting, as they have the strongest contact with the patients and are overrepresented in the survey with almost 60%.

Thank you very much for this valid point. We did subgroup analyses regarding gender (males/females) and status group (nurses/other) but were rather conservative with explorative explanations as this marked the first time of this kind of study being conducted at KHI. In alignment with other studies like SeViD-I and SeViD-II studies comparing doctors and nurses in Germany we did not show any significant deviations concerning symptom load or prevalence for gender or status group. In order to add this information that indeed was not presented before we amended the manuscript where appropriate to indicate the lack of substantial deviations.

Also findings from our subgroup analysis about status groups (nurses/others) have been added to the text and later discussed. Finally, as nurses are the largest occupational group at KHI representing a comparable proportion of employees in our sample acknowledge them to be the most prominent status group within the hospital and our survey, but we don’t consider them to be over-represented.

Conclusion

What can the authors recommend after finalizing this study? Which targeted strategies, as stated in the introduction (L. 43-44) should be developed? What about the result from L. 210 ‘1. The possibility to access legal consultation after a severe event’?

The results of this study justify the implementation of the KoHi program. This program will be based on other, established second victim response programs, such as RISE or ForYOU. We added this to the results section, so it now reads:

“The results of this study justify the implementation of the KoHi program. The program is based on other, established Second Victim support programs, such as RISE at Johns Hopkins Hospital [24] or forYOU at University of Missouri Health Care [15].”

  1. 234-235 What does KoHi mean?

It stands for “Kollegiale Hilfe” (Collegial Help) and is the name of the study as well as the support program implemented at KHI to tackle the SVP. We added this description in the introduction. Thank you for pointing this out, as it adds to the understandability of our manuscript.

Thank you very much!

Thank you very much for your valuable comments that helped to improve our paper.

***end of review***

Reviewer 3 Report

This manuscript reports the results of a survey in a German healthcare facility with the main purpose of reporting the prevalence of the Second Victim Phenomenon (SVP), which is being traumatized by medical errors and adverse patient outcomes.

I thought the authors did a good job of defining the SVP, explaining why it was important, and also why more research is needed on its prevalence.  With that being said, I do have some serious concern with the manuscript, and these are listed below:

1.  While I do think there is value in prevalence studies in occupational health, I'm not sure this study is design in a way that sheds much light on the overall prevalence of SVP.  Since it is done in one facility, and participants are not chosen based on any type of probability sampling, I'm not what the results tell us about levels of SVP beyond the setting in which it was conducted.

2.  At one point in the manuscript the authors discuss qualitative findings (Lines 85-92), but don't really elaborate on them very much.  Perhaps a more in-depth discussion of these qualitative findings could be used to explain the quantitative results.

3.  The authors present a rather long list of reactions to SVP in Table 4, and some of these are interesting.  I would suggest trying to relate these to the form of SVP experienced.

4.  While I do agree with the authors that the response rate for the study was not bad, I would still like to see some attempt to see how representative the sample was (e.g., compare to the overall demographics of the employees in the facility).

Author Response

Reviewer 3

This manuscript reports the results of a survey in a German healthcare facility with the main purpose of reporting the prevalence of the Second Victim Phenomenon (SVP), which is being traumatized by medical errors and adverse patient outcomes.

I thought the authors did a good job of defining the SVP, explaining why it was important, and also why more research is needed on its prevalence.  With that being said, I do have some serious concern with the manuscript, and these are listed below:

  1. While I do think there is value in prevalence studies in occupational health, I'm not sure this study is design in a way that sheds much light on the overall prevalence of SVP.  Since it is done in one facility, and participants are not chosen based on any type of probability sampling, I'm not what the results tell us about levels of SVP beyond the setting in which it was conducted.

We agree to your valid question that monocentric studies have to justify their relevance for readers. Concerning Second Victim Phenomenon in German speaking countries there is a relevant taboo to even acknowledge the presence of this phenomenon in general. That is why the first surveys in Germany were conducted under the auspices of medical and/or professional associations. To our knowledge this is the first surveys in a single institution in German-speaking hospital that is about to be published. It has not only potential to help overcoming a widespread taboo but may also serve as single-center reference for all other hospitals in German-speaking countries. As KoHi I is part of multi-stepwise systematic approach to prospectively plan and conduct a Second Victim support program we also consider this study to serve as a blueprint for other institutions that aim to implement such measures.

  1. At one point in the manuscript the authors discuss qualitative findings (Lines 85-92), but don't really elaborate on them very much.  Perhaps a more in-depth discussion of these qualitative findings could be used to explain the quantitative results.

Within the questionnaire there was only one small field to provide free-text regarding the cause of Second Victim traumatization for validation purposes and to demonstrate variety of causes of traumatization by giving a few examples. Therefore, we don´t consider these answers to be valid for deeper analysis. In order to clarify this we have added additional information about the questionnaire in the methods section

  1. The authors present a rather long list of reactions to SVP in Table 4, and some of these are interesting.  I would suggest trying to relate these to the form of SVP experienced.

Thank you for this suggestion. The list of this content-validated questionnaire refers to previous studies conducted in English speaking countries. We thought about your suggestion but were not able to detect any specific pattern concerning relation of harm and specific reactions. Maybe large-scale studies can provide such information in the future.

  1. While I do agree with the authors that the response rate for the study was not bad, I would still like to see some attempt to see how representative the sample was (e.g., compare to the overall demographics of the employees in the facility).

We absolutely agree to your request to verify internal validity of our data. While we see the relation of gender and status groups (e.g. doctors, nurses etc.) to be quite similar to the distribution of overall staff of KHI presentation of any in depth analysis would be regarded as a potential violation against data protection agreements since linking gender, status group and duration of occupation and KHI will bear the risk of unintentional identification of individuals in status group combinations with low samples. Therefore, we explicitly refrained from giving detailed numbers of staff of KHI, but added the information that demographics of gender and status groups are comparable to overall demographics of KHI in the discussion section.

Round 2

Reviewer 1 Report

I am pleased with the authors' response.

Author Response

Dear Reviewer,

Thank you very much once again for your kind support and your valuable comments.